# Towards Understanding the Role of Adaptive Learning Rates in Powered Stochastic Gradient Descent

## Abstract

The use of the adaptive learning rate (ALR) in stochastic gradient-based methods has become a wide practice in machine learning, even become a default mode in training deep learning. Different variants of ALR techniques, including AdaGrad, Adam, AMSGRAD, and RMSProp, have reported significant success in improving stochastic optimization. Despite these empirical successes, there is an extremely lack of clear comprehending how various ALR techniques affect both theoretical and empirical behaviors of powered stochastic gradient-based algorithms. Even, the impact of existing ALR techniques in common stochastic gradient-based algorithms is still under-explored. To fill the gap, this work develops a novel powered stochastic gradient-based algorithm with generalized adaptive learning rates, coined ADAptive Powered Stochastic Gradient Descent (ADA-PSGD), for nonconvex optimization problems. We particularly elucidate numerous connections of ADA-PSGD to existing ALR techniques. Moreover, we prove a faster convergence rate of ADA-PSGD for nonconvex optimization problems. Further, we show that ADA-PSGD achieves a gradient evaluation cost of $O\left(n + L^2\|\mathbf{1}\|_p^2(1 - \alpha_1\beta_1)^{-1}\varepsilon^{-2}\right)$ ($\alpha_1 \in [0,1]$ and $\beta_1 \in [0,1)$) to find an $\varepsilon$-approximate stationary point, which is comparable to the well-known algorithmic lower bound. Finally, we empirically demonstrate that our ADA-PSGD algorithm leads to greatly improved training in different machine learning tasks. Further, we hope that the robustness of ADA-PSGD to crucial hyper-parameters will spur interest from both researchers and practitioners.

## 1 Introduction

In order to solve many applications of science and engineering, what are the generations of researchers have been hard working is fast and robust optimization algorithms. Stochastic optimization, such as stochastic gradient descent (SGD), has reported tremendous success in many areas of machine learning (Liu et al., 2024), computer vision (Croitoru et al., 2023), management science (Chen et al., 2024), etc., although its operation is much simplicity. During recent years, many efforts and attempts have been proposed to modifying stochastic optimization, including but not limited to (1) high-order information (Cao et al., 2023); (2) conjugate gradient (Yang, 2023b); (3) Powerball techniques (Zhou et al., 2021); (4) momentum (Loizou & Richtárik, 2020); (5) adaptive learning rates (Zhang et al., 2022); (6) Polyak learning rate (Orvieto et al., 2022); (7) importance sampling (Zhao & Zhang, 2015); (8) mini-batching sampling (Cotter et al., 2011); (9) shuffling strategy (Malinovsky et al., 2023). Actually, most of the existing techniques are contributed to utilize the way of renewing gradient direction (e.g., (1)-(4)), or updating the learning rate (e.g., (5), (6)) to accelerate stochastic optimization methods. This work mainly focuses on the development of Powerball techniques and adaptive learning rates in improving stochastic optimization, referring two main guidelines to advance the progress of stochastic optimization currently.

Adaptive learning rates, also familiar with adaptive gradient methods, often acquire the learning rate by the form of exponentially decaying average of squared historical gradient values. Specifically, AdaGrad (Duchi et al., 2011) and its variants, such as RMSProp (Tieleman et al., 2012), Adam (Kingma, 2014), AMSGRAD (Reddi et al., 2018), etc., are representative because of their fast training speed. Although prevailing, adaptive gradient methods are observed to generalize poorly in contrast to plain SGD, or even fail to converge even

under well-defined problems (e.g., convex case). Then a large number of existing studies are conducted to establish the convergence guarantee of adaptive gradient methods. For example, originally, Kingma (2014) analyzed that Adam with decreasing effective learning rates converged to an optimal solutions for convex case. Nevertheless, the mistake of the proof in (Kingma, 2014) was pointed out by Reddi et al. (2018), who also exemplified the divergence of Adam on a simple convex problem. Further, Reddi et al. (2018) provided one reason that adaptive gradient methods diverge practically is the exponential moving average adopted in the algorithms and showed that the introduction of long-term memory of previous gradients into adaptive gradient methods can fix this issue, leading to AMSGRAD. Considering the techniques in (Ward et al., 2020) and extending them to the Adam optimizer, Défossez et al. (2022) offered a simple proof of the convergence of Adam and AdaGrad under the assumptions of smooth gradients and almost sure uniform bound on the $\ell_\infty$ norm of the gradients. More information about adaptive gradient methods is discussed in Section 2.

In contrast, Powerball methods (Zhou et al., 2021; Yuan et al., 2019) improve stochastic optimization by adding a power coefficient $\gamma \in [0, 1)$ to the gradient, which is orthogonal to the adaptive gradient methods in accelerating stochastic optimization. More generally, we can regard the Powerball method as the steepest gradient descent approach with respect to the $p$-norm, where $p = 1 + (1/\gamma)$. Interestingly, Newton-type methods, employing curvature information of the model, usually converge faster than the steepest gradient descent and conjugate gradient methods. More interestingly, Newton-type algorithms are also viewed as the steepest gradient descent methods with the ellipsoidal norm. This also demonstrates that why the Powerball technique can improve stochastic optimization from the side. Zhou et al. (2021) applied the Powerball technique into SGD and SGD with momentum (SGDM), respectively, leading to pbSGD and pbSGDM, and proved that the convergence rates of pbSGD and pbSGDM is comparable to well-known convergence rates for SGD and SGDM on nonconvex functions. Yang (2023a) further improved powered stochastic optimization by utilizing the advantage of variance-reduced technique and showed that their proposed PB-SVRGE method converged with rate $O(\frac{1}{\sqrt{1+2BT}})$, where $B$ represented batch sample per iterative step and $T$ denotes the total number of iterations.

**Motivation.** Powered stochastic optimization algorithms are gradient modifiers via a nonlinear transformation, which is totally different from the mechanism of adaptive learning rates in accelerating stochastic optimization algorithms. However, very usual, the selection of learning rates for powered stochastic optimization algorithms is more challenging than SGD due to the nonlinear transformation and is lacking of studies to solve this issue to date. On the other side, although this work is inspired by adaptive gradient methods, there exist too many different adaptive gradient methods at present. Therefore, one is greatly desirable to know that which certain type of adaptive gradient methods are the best for enhancing powered stochastic optimization algorithms. More generally, it is suspicious that whether a unified analysis of powered stochastic optimization algorithms with various adaptive gradient methods can be established.

The above discussions further inspire us to come up with the following research questions:

1) *Can we utilize both the Powerball technique and the rule of adaptive learning rates to develop new provable, fast and robust stochastic gradient-based algorithms for practical applications of science and engineering?*

2) *Naturally, for various existing adaptive learning rates, e.g., AdaGrad, Adam, RMSProp, AMSGRAD, etc., which one is more suitable for the powered stochastic optimization algorithms?*

3) *Further, does there exist a unified analysis for powered stochastic optimization algorithms with different adaptive learning rates?*

In this work, we answer these questions by developing a new algorithm, coined ADAptive Powered Stochastic Gradient Descent (ADA-PSGD). We provide its theoretical guarantees for different adaptive learning rates. More specifically, a unified convergence analysis of ADA-PSGD under different adaptive learning rates is provided.

**Our contribution.** At last, we summarize our contributions in this work as follows:

(1) We propose a new powered stochastic gradient-based algorithm with generalized adaptive learning rates (Algorithm 3 in Section 4) for solving the nonconvex optimization problem. Our algorithm contains most of existing adaptive learning rates, e.g., RMSProp, Adam, NAdam (Dozat, 2016).

(2) We establish a unified convergence analysis of our algorithm with different adaptive learning rates in the nonconvex optimization background under standard assumptions (a.k.a., the $L$-Lipschitz smooth gradient and bounded variance conditions).

(3) We prove that our algorithm obtains gradient computation cost of $O\left(n + L^2\|\mathbf{1}\|_p^2(1 - \alpha_1\beta_1)^{-1}\varepsilon^{-2}\right)$ ($\alpha_1 \in [0,1]$ and $\beta_1 \in [0,1)$) to find an $\varepsilon$-approximate stationary point, which matches the algorithmic lower bound, attaining by modern stochastic gradient-based algorithms, e.g., SVRG (Johnson & Zhang, 2013), SARAH (Nguyen et al., 2017), SPIDER (Fang et al., 2018), etc.

(4) We, by comparing state-of-the-art stochastic gradient-based algorithms, empirically demonstrate that our algorithm leads to significantly improved training in various machine learning tasks. Moreover, tons of numerical experiments verify the robustness of ADA-PSGD to crucial hyper-parameters.

## 2 Related Work

Most of the existing studies pay much attention to how the Powerball function improves stochastic optimization algorithms, but do not care about the effect of the learning rate in powered stochastic optimization algorithms. For instance, Yuan et al. (2019) showed a positive effect of the Powerball technique in accelerating both (stochastic) gradient-based and limited-memory Broyden-Fletcher-Goldfarb-Shanno (L-BFGS) approaches. Zhang & Bailey (2022) demonstrated the potential of the Powerball technique in improving the zeroth-order distributed primal-dual stochastic coordinate method. More recently, we found that Yang (2023a) proposed using the Barzilai-Borwein-like technique to compute the learning rate for the powered stochastic gradient-based algorithm. Yang & Li (2024) applied the hypergradient descent technique into the powered stochastic gradient-based algorithm to acquire the learning rate.

In addition, whereas adaptive gradient methods are significantly popular in well-studied stochastic gradient-based algorithms, most of the existing studies usually explores one of them for stochastic gradient-based algorithms. Several recent works attempt to offer a unified perspective of stochastic gradient-based algorithms with different adaptive gradient methods. Xiao et al. (2024) provided a comprehensive research on the convergence of Adam-family approaches by developing a new two-timescale framework for nonsmooth optimization, where their proposed framework contained various preferred Adam-family methods (including Adam, AMSGRAD, etc). By equipping a general form of the second-order moment, Jiang et al. (2023) introduced and analyzed a unified framework for Adam-type algorithms (referred to as UAdam) for nonconvex stochastic setting, where UAdam included NAdam, AMSGRAD, AdaBound (Luo et al., 2019), etc.

## 3 Preliminaries

Many practical applications in computer vision, machine learning, and natural language processing are attributed to addressing the following composite minimization:

$$\min_{x\in\mathbb{R}^d} F(x) = \frac{1}{n}\sum_{i=1}^n f_i(x). \tag{1}$$

where each $f_i(x) : \mathbb{R}^d \to \mathbb{R}$ usually denotes a smooth and possibly nonconvex function for $i \in [n] := \{1, \cdots, n\}$.

Throughout the paper, let $\|x\|$ and $\|x\|_p$ denote the Euclidean norm and $\ell_p$-norm of $x$, respectively. For vector $x$, we employ $(x)_i$ to denote its $i$-th coordinate. We use $\nabla F(w)$ to denote the gradient of $F(w)$. We denote $[n] = \{1, 2, \ldots, n\}$. We write $\langle x, y \rangle$ as the inner product of vectors $x$ and $y$. We write $\mathbb{E}[z]$ as the expectation of a random variable $z$. For any given two sequences, $\{a_n\}$ and $\{b_n\}$, we denote $a_n = O(b_n)$, if there exists a constant $C > 0$ such that $a_n \leq Cb_n$.

**Generic adaptive gradient methods.** Algorithm 1 presents a generic framework of adaptive gradient methods. Various popular stochastic optimization algorithms can be shown by Algorithm 1 through specified different choice of $\phi_t$ and $\psi_t$, where $\phi_t$ offers how the momentum term at $t$-th timestep is computed, and $\psi_t$ offers how the adaptive learning rate at $t$-th timestep is computed. For instance, in Adam, we have

$$\textbf{Adam} : \begin{cases} \phi_t(g_1,\ldots,g_t) = (1-\beta_1)\sum_{i=1}^t \beta_1^{t-i} g_i, \\ \psi_t(g_1,\ldots,g_t) = (1-\beta_2)\sum_{i=1}^t \beta_2^{t-i} g_i^2, \end{cases} \tag{2}$$

where $\phi_t(g_1,\ldots,g_t)$ is derived from $m_t = \beta_1 m_{t-1} + (1-\beta_1)g_t$ and $\psi_t(g_1,\ldots,g_t)$ is derived from $l_t = \beta_2 v_{t-1} + (1-\beta_2)g_t^2$. Two parameters $\beta_1$ and $\beta_2$ are belonging to $(0,1)$.

In contrast, in RMSProp, we have

$$\textbf{RMSProp} : \begin{cases} \phi_t(g_1,\ldots,g_t) = g_t, \\ \psi_t(g_1,\ldots,g_t) = (1-\beta_2)\sum_{i=1}^t \beta_2^{t-i} g_i^2, \end{cases} \tag{3}$$

---

**Algorithm 1** Generic adaptive gradient method setup

**Require:** : $\{\alpha_t\}_{t=1}^T$: step size, $\{\phi_t, \psi_t\}_{t=1}^T$: function to evaluate momentum and adaptive learning rate, $\epsilon \in (0,1)$.
1: **for** $t = 1$ **to** $T$ **do**
2:     $g_t = \nabla f_{n_t}(x_t)$ (Compute gradients at $t$-th timestep)
3:     $m_t = \phi_t(g_1,\ldots,g_t)$ (Compute momentum)
4:     $l_t = \psi_t(g_1,\ldots,g_t)$ (Compute adaptive learning rate)
5:     $x_{t+1} = x_t - \alpha_t m_t/(\sqrt{l_t} + \epsilon)$ (Renew parameters)
6: **end for**

---

**Powered stochastic optimization algorithms.** Algorithm 2 is the framework of powered stochastic optimization algorithms. In Algorithm 2, the mapping $\sigma_\gamma$, named the Powerball function, is applied to each element of $g_t$, that is $\sigma_\gamma(g_t) = (\sigma_\gamma((g_t)_1), \sigma_\gamma(g_t)_2), \cdots, \sigma_\gamma((g_t)_d))^T$. The Powerball function $\sigma(\cdot) : \mathbb{R} \to \mathbb{R}$ is of the form $\sigma_\gamma(a) = \text{sign}(a)|a|^\gamma$ for $\gamma \in [0,1)$, where $\text{sign}(a)$ goes to the sign of $a$ if $a \neq 0$, or 0 if $a = 0$. In particular, for $\gamma = 1$, Algorithm 2 reduces to plain SGD, while for $\gamma = 0$, Algorithm 2 goes to SIGNSGD (Bernstein et al., 2018).

---

**Algorithm 2** Powered stochastic optimization algorithm setup

**Require:** : $\{\alpha_t\}_{t=1}^T$: step size, $\gamma \in [0,1)$: power coefficient
1: **for** $t = 1$ **to** $T$ **do**
2:     $g_t = \nabla f_{n_t}(x_t)$ (Compute gradients at $t$-th timestep)
3:     $x_{t+1} = x_t - \alpha_t \sigma_\gamma(g_t)$ (Renew parameters)
4: **end for**

---

## 4 The Algorithm

We equip powered stochastic optimization algorithms with generalized adaptive learning rates by introducing the quasi-hyperbolic momentum (QHM) into adaptive gradient methods, where QHM solves Problem (1) with the following update scheme (Ma & Yarats, 2019):

$$\textbf{QHM} : \begin{cases} m_t = \beta m_{t-1} + (1-\beta)g_t, \\ x_t = x_{t-1} - \eta[\chi m_t + (1-\chi)g_t], \end{cases} \tag{4}$$

where $\chi \in [0,1]$. Following will show the connections of QHM with several existing momentum methods.

Actually, Adam-family methods often adopt the heavy-ball momentum (HBM) (Li et al., 2021a) whose update scheme is

$$\textbf{HBM} : \begin{cases} m_t = \beta m_{t-1} + (1-\beta)g_t, \\ x_t = x_{t-1} - \eta m_t. \end{cases} \tag{5}$$

Different from the original HBM method, Adam-family methods is normalized by $1 - \beta$, which not only alleviates dependence of the update step magnitude on the momentum coefficient $\beta$, but permits the interpretation of $m_t$ as a weighted average of previous gradients. Under the case $\chi = 1$, QHM is reduced to HBM. In contrast, under the case $\chi = 0$, QHM is reduced to plain SGD.

Several studies (Zhou et al., 2022; Chen et al., 2022) accelerate adaptive gradient methods from the perspective of Nesterov's accelerated gradient (NAG), whose update scheme is usually formulated as (Nesterov, 2013)

$$
\textbf{NAG} : \left\{
\begin{array}{l}
g_t = \nabla F_{n_t}(x_t - \alpha(1 - \beta)m_{t-1}), \\
m_t = \beta m_{t-1} + g_t, \\
x_t = x_{t-1} - \eta m_t,
\end{array}
\right.
\tag{6}
$$

where NAG updates parameters by using the gradient at the extrapolation point $x'_t = x_t - (1 - \beta)(x_t - x_{t-1})$ that is different from HBM. More specifically, setting $\chi = \beta$, QHM turns out to be a normalized variant of NAG with an additional factor $1 - \beta$.

Now, we are ready to describe our algorithm, referred to as ADA-PSGD, in Algorithm 3.

---

**Algorithm 3** ADA-PSGD

---

**Require:** base learning rate $\eta$, outer loop size $\mathfrak{S}$, batch sample $B$, the preconditioned parameters $\epsilon$, $\eta$, $\gamma$, $\alpha_1$, $\alpha_2$, $\beta_1$, and $\beta_2$

    **Initialize:** $\widetilde{x}^0$, $G_0^0 = \mathbf{0}$, and $U_0^0 = \mathbf{0}$

    **for** $s = 1$ **to** $\mathfrak{S}$ **do**

        $\tilde{x} = x_0^s = \widetilde{x}^{s-1}$

        $g^s = \nabla F(\tilde{x})$

        **for** $k = 1$ **to** $\mathfrak{K}$ **do**

            Select random mini-batch $n_k$ from $[n]$ with $B$ samples

        $V_k^s = \nabla F_{n_k}(x_{k-1}^s) - \nabla F_{n_k}(\tilde{x}) + g^s$

        $G_k^s = \beta_1 G_{k-1}^s + (1 - \beta_1) V_k^s$

        $U_k^s = \beta_2 U_{k-1}^s + (1 - \beta_2)(\nabla F_{n_k}(x_{k-1}^s))^2$

        $x_k^s = x_{k-1}^s - \eta \left[ \dfrac{\sigma_\gamma(\alpha_1 G_k^s + (1 - \alpha_1) V_k^s)}{\sqrt{\alpha_2 U_k^s + (1 - \alpha_2)(\nabla F_{n_k}(x_{k-1}^s))^2} + \epsilon} \right]$

        **end for**

        $\widetilde{x}^s = x_{\mathfrak{K}}^s$

    **end for**

---

**Remark:** To know ADA-PSGD well, some explanations for ADA-PSGD are presented here.

(1) We first show the connections between ADA-PSGD and existing adaptive gradient methods. Specifically, under the case $\gamma = 1$, ADA-PSGD (i) is reduced to the Adam-like algorithm when $\alpha_1 = \alpha_2 = 1$; (ii) turns out to the RMSProp-like algorithm when $\alpha_1 = 0$ and $\alpha_2 = 1$; (iii) goes to the NAdam when $\alpha_1 = \beta_1$ and $\alpha_2 = 1$. More generally, this work extends the family of adaptive gradient methods by developing ADA-PSGD with $\gamma \in (0, 1)$.

(2) Further, ADA-PSGD uses the stochastic variance-reduced gradient (SVRG) estimator, i.e., $V_k^s = \nabla F_{n_k}(x_{k-1}^s) - \nabla F_{n_k}(\tilde{x}) + g^s$, where the SVRG-gradient estimator is also considered in (Dubois-Taine et al., 2022). Nevertheless, Dubois-Taine et al. (2022) only considers the case of AdaGrad with the SVRG-gradient estimator and without the Powerball function. Additionally, we also find that Kavis et al. (2022) proposed ADASPIDER by introducing the SPIDER gradient estimator into AdaGrad.

## 5 Convergence Analysis

This section provides a theoretical guarantee of ADA-PSGD for the nonconvex optimization problem. The following common assumptions in analyzing stochastic optimization algorithms are required.

**Assumption 1.** *The following conditions are kept for the objective function, $F(x)$ and its gradient, in Problem (1):*

(a) *The objective function, $F(x)$, has L-Lipschitz smooth gradient, i.e., there exists a positive constant $L$ such that for $\forall x, y \in \mathbb{R}^d$*

$$\|\nabla F(y) - \nabla F(x)\| \leq L\|y - x\| \tag{7}$$

(b) *$\|\nabla F_{n_k}(x) - \nabla F(x)\|^2$ has an upper boundary, i.e.,*

$$\mathbb{E}[\|\nabla F_{n_k}(x) - \nabla F(x)\|^2] \leq \frac{\sigma^2}{B}, \tag{8}$$

*where, $n_k \subset [n]$, with $B$ samples.*

From the result of Assumption 1(a), the following conclusion for $L$-smooth function is directly derived:

**Lemma 1. (Descent Lemma)** *Assumption 1(a) indicates that there exists a quadratic upper bound on $F$*

$$F(y) \leq F(x) + \langle \nabla F(x), y - x \rangle + \frac{L}{2}\|y - x\|^2. \tag{9}$$

Ghadimi & Lan (2013) proved Lemma 1 by utilizing the Taylor expansion of the function $F(x)$ around $x$. This lemma can also be found in (Nesterov, 2013).

To proceed analysis of Theorem 1, the following two lemmas are required.

**Lemma 2.** *For $\alpha_1 G_k^s + (1 - \alpha_1)V_k^s$, generated by Ada-PSGD, we have the following boundary:*

$$\|\alpha_1 G_k^s + (1 - \alpha_1)V_k^s\|_{1+\gamma}^2 \geq \frac{1}{2}(1 - \alpha_1\beta_1)^2\|\nabla F_{n_k}(x_{k-1}^s)\|_{1+\gamma}^2 - \|(1 - \alpha_1\beta_1)\nabla F_{n_k}(\tilde{x}) - \alpha_1\beta_1 G_{k-1}^s$$
$$- (1 - \alpha_1\beta_1)g^s\|_{1+\gamma}^2. \tag{10}$$

*Proof.* Since $G_k^s = \beta_1 G_{k-1}^s + (1 - \beta_1)V_k^s$ and $V_k^s = \nabla F_{n_k}(x_{k-1}^s) - \nabla F_{n_k}(\tilde{x}) + g^s$, defined in Algorithm 3, we easily ascertain

$$\|\alpha_1 G_k^s + (1 - \alpha_1)V_k^s\|_{1+\gamma}^2 = \|\alpha_1\beta_1 G_{k-1}^s + \alpha_1(1 - \beta_1)V_k^s + (1 - \alpha_1)V_k^s\|_{1+\gamma}^2$$
$$= \|\alpha_1\beta_1 G_{k-1}^s + (1 - \alpha_1\beta_1)V_k^s\|_{1+\gamma}^2$$
$$= \|\alpha_1\beta_1 G_{k-1}^s + (1 - \alpha_1\beta_1)\nabla F_{n_k}(x_{k-1}^s) - (1 - \alpha_1\beta_1)\nabla F_{n_k}(\tilde{x}) + (1 - \alpha_1\beta_1)g^s\|_{1+\gamma}^2$$
$$\geq \frac{1}{2}(1 - \alpha_1\beta_1)^2\|\nabla F_{n_k}(x_{k-1}^s)\|_{1+\gamma}^2 - \|(1 - \alpha_1\beta_1)\nabla F_{n_k}(\tilde{x}) - \alpha_1\beta_1 G_{k-1}^s$$
$$- (1 - \alpha_1\beta_1)g^s\|_{1+\gamma}^2, \tag{11}$$

where the first inequality uses the condition $\|a\|_{1+\gamma}^2 \geq \frac{1}{2}\|b\|_{1+\gamma}^2 - \|b - a\|_{1+\gamma}^2$. $\square$

**Lemma 3.** *Under Assumption 1(b), for $\alpha_1 G_k^s + (1 - \alpha_1)V_k^s - \nabla F(x_{k-1}^s)$, we get the following upper boundary*

$$\|\alpha_1 G_k^s + (1 - \alpha_1)V_k^s - \nabla F(x_{k-1}^s)\|^2 \leq 4\alpha_1^2\beta_1^2\|G_{k-1}^s\|^2 + \frac{8\sigma^2(1 - \alpha_1\beta_1)^2}{B} + 4\alpha_1^2\beta_1^2\|\nabla F(x_{k-1}^s)\|^2. \tag{12}$$

*Proof.*

$$\|\alpha_1 G_k^s + (1 - \alpha_1)V_k^s - \nabla F(x_{k-1}^s)\|^2$$
$$= \|\alpha_1\beta_1 G_{k-1}^s + (1 - \alpha_1\beta_1)[\nabla F_{n_k}(x_{k-1}^s) - \nabla F_{n_k}(\tilde{x}) + g^s] - \nabla F(x_{k-1}^s)\|^2$$
$$= \|\alpha_1\beta_1 G_{k-1}^s + (1 - \alpha_1\beta_1)\nabla F_{n_k}(x_{k-1}^s) - (1 - \alpha_1\beta_1)\nabla F(\tilde{x})$$
$$- (1 - \alpha_1\beta_1)\nabla F_{n_k}(\tilde{x}) + (1 - \alpha_1\beta_1)\nabla F(\tilde{x}) - \alpha_1\beta_1\nabla F(x_{k-1}^s)\|^2$$
$$\leq 4\alpha_1^2\beta_1^2\|G_{k-1}^s\|^2 + 4(1 - \alpha_1\beta_1)^2\|\nabla F_{n_k}(x_{k-1}^s) - \nabla F(x_{k-1}^s)\|^2$$
$$+ 4(1 - \alpha_1\beta_1)^2\|\nabla F_{n_k}(\tilde{x}) - \nabla F(\tilde{x})\|^2 + 4\alpha_1^2\beta_1^2\|\nabla F(x_{k-1}^s)\|^2$$
$$\leq 4\alpha_1^2\beta_1^2\|G_{k-1}^s\|^2 + \frac{8\sigma^2(1 - \alpha_1\beta_1)^2}{B} + 4\alpha_1^2\beta_1^2\|\nabla F(x_{k-1}^s)\|^2, \tag{13}$$

where the first inequality uses the fact $(a_1 + a_2 + \ldots + a_n)^2 \leq n(a^2 + a^2 + \ldots + a_n^2)$ and the second inequality uses Assumption 1(b).

$\square$

The main theoretical result of ADA-PSGD is given in Theorem 1.

**Theorem 1.** *Suppose Assumption 1, Lemma 1, Lemma 2 and Lemma 3 hold and choose $n_k \subseteq [n]$ with $B$ samples. Let $x_* = \arg\min_{x \in \mathbb{R}^d} F(x)$ and $\{x_k^s\}_{k=1,s=1}^{\mathfrak{K},\mathfrak{G}}$ generated by ADA-PSGD. For any $T \geq 1$, ADA-PSGD leads to*

$$\mathbb{E}\left[\frac{1}{T}\sum_{s=1}^{\mathfrak{G}}\sum_{k=1}^{\mathfrak{K}}\|\nabla F_{n_k}(x_{k-1}^s)\|_{1+\gamma}^2\right] \leq \frac{4L\|\mathbf{1}\|_p}{(1-\theta)T(1-\alpha_1\beta_1)^2}[F(\tilde{x}_0) - F(x_*)] + \frac{16\|\mathbf{1}\|_p\sigma^2}{(1-\theta)B\theta}, \tag{14}$$

*where $\theta > 0$.*

*Proof.* Based on the $L$-smooth property of the loss function, $F(w)$, and the definition of $x_k^s$ that is $x_k^s = x_{k-1}^s - \eta\left[\frac{\sigma_\gamma(\alpha_1 G_k^s + (1-\alpha_1)V_k^s)}{\sqrt{\alpha_2 U_k^s + (1-\alpha_2)(\nabla F_{n_k}(x_{k-1}^s))^2 + \epsilon}}\right]$, defined in Algorithm 3, we have

$$\mathbb{E}[F(x_k^s)] \leq \mathbb{E}\left[F(x_{k-1}^s) + \langle \nabla F(x_{k-1}^s), x_k^s - x_{k-1}^s \rangle + \frac{L}{2}\|x_k^s - x_{k-1}^s\|^2\right]$$

$$= \mathbb{E}\left[F(x_{k-1}^s) - \eta\left\langle \nabla F(x_{k-1}^s), \frac{\sigma_\gamma(\alpha_1 G_k^s + (1-\alpha_1)V_k^s)}{\sqrt{\alpha_2 U_k^s + (1-\alpha_2)(\nabla F_{n_k}(x_{k-1}))^2 + \epsilon}}\right\rangle\right.$$

$$+ \left.\frac{L\eta^2}{2}\left\|\frac{\sigma_\gamma(\alpha_1 G_k^s + (1-\alpha_1)V_k^s)}{\sqrt{\alpha_2 U_k^s + (1-\alpha_2)(\nabla F_{n_k}(x_{k-1}))^2 + \epsilon}}\right\|^2\right]$$

$$= \mathbb{E}\left[F(x_{k-1}^s) - \eta\left\langle \alpha_1 G_k^s + (1-\alpha_1)V_k^s, \frac{\sigma_\gamma(\alpha_1 G_k^s + (1-\alpha_1)V_k^s)}{\sqrt{\alpha_2 U_k^s + (1-\alpha_2)(\nabla F_{n_k}(x_{k-1}))^2 + \epsilon}}\right\rangle + \eta\right.$$

$$\cdot \left\langle \alpha_1 G_k^s + (1-\alpha_1)V_k^s - \nabla F(x_{k-1}^s), \frac{\sigma_\gamma(\alpha_1 G_k^s + (1-\alpha_1)V_k^s)}{\sqrt{\alpha_2 U_k^s + (1-\alpha_2)(\nabla F_{n_k}(x_{k-1}))^2 + \epsilon}}\right\rangle\left. + \frac{L\eta^2}{2}\right.$$

$$\cdot \left.\left\|\frac{\sigma_\gamma(\alpha_1 G_k^s + (1-\alpha_1)V_k^s)}{\sqrt{\alpha_2 U_k^s + (1-\alpha_2)(\nabla F_{n_k}(x_{k-1}))^2 + \epsilon}}\right\|^2\right]. \tag{15}$$

For convenience, in the following, the token, $\Diamond$, is used to represent $\alpha_1 G_k^s + (1-\alpha_1)V_k^s$ and the token, $\triangle$, is used to represent $\sqrt{\alpha_2 U_k^s + (1-\alpha_2)(\nabla F_{n_k}(x_{k-1}))^2} + \epsilon$. Considering $\eta = \left\langle \Diamond, \frac{\sigma_\gamma(\Diamond)}{\triangle}\right\rangle \Big/ L\left\|\frac{\sigma_\gamma(\Diamond)}{\triangle}\right\|^2 > 0$, we have

$$\mathbb{E}[F(x_k^s)] \leq \mathbb{E}\left[F(x_{k-1}^s) - \left(\left\langle \Diamond, \frac{\sigma_\gamma(\Diamond)}{\triangle}\right\rangle \Big/ L\left\|\frac{\sigma_\gamma(\Diamond)}{\triangle}\right\|^2\right)\cdot\left\langle \Diamond, \frac{\sigma_\gamma(\Diamond)}{\triangle}\right\rangle + \frac{L}{2}\left(\left\langle \Diamond, \frac{\sigma_\gamma(\Diamond)}{\triangle}\right\rangle \Big/ L\left\|\frac{\sigma_\gamma(\Diamond)}{\triangle}\right\|^2\right)^2\right.$$

$$\cdot \left\|\frac{\sigma_\gamma(\Diamond)}{\triangle}\right\|^2 + \left(\left\langle \Diamond, \frac{\sigma_\gamma(\Diamond)}{\triangle}\right\rangle \Big/ L\left\|\frac{\sigma_\gamma(\Diamond)}{\triangle}\right\|^2\right)\left\langle \Diamond - \nabla F(x_{k-1}^s), \frac{\sigma_\gamma(\Diamond)}{\triangle}\right\rangle\Bigg]$$

$$= \mathbb{E}\left[F(x_{k-1}^s) - \left(\left\langle \Diamond, \frac{\sigma_\gamma(\Diamond)}{\triangle}\right\rangle\right)^2 \Big/ L\left\|\frac{\sigma_\gamma(\Diamond)}{\triangle}\right\|^2 + \frac{1}{2L}\left(\left\langle \Diamond, \frac{\sigma_\gamma(\Diamond)}{\triangle}\right\rangle\right)^2 \Big/ L\left\|\frac{\sigma_\gamma(\Diamond)}{\triangle}\right\|^2\right.$$

$$+ \left.\left(\left\langle \Diamond, \frac{\sigma_\gamma(\Diamond)}{\triangle}\right\rangle \Big/ L\left\|\frac{\sigma_\gamma(\Diamond)}{\triangle}\right\|^2\right)\left\langle \Diamond - \nabla F(x_{k-1}^s), \frac{\sigma_\gamma(\Diamond)}{\triangle}\right\rangle\right]. \tag{16}$$

Rearranging the inequality (16), we further have

$$
\begin{aligned}
\mathbb{E}[F(x_k^s)] \leq{} & \mathbb{E}\bigg[F(x_{k-1}^s) - \frac{1}{2L}\bigg(\bigg\langle \Diamond, \frac{\sigma_\gamma(\Diamond)}{\Delta}\bigg\rangle\bigg)^2 \bigg/ L\bigg\|\frac{\sigma_\gamma(\Diamond)}{\Delta}\bigg\|^2 + \bigg(\bigg\langle \Diamond, \frac{\sigma_\gamma(\Diamond)}{\Delta}\bigg\rangle \bigg/ L\bigg\|\frac{\sigma_\gamma(\Diamond)}{\Delta}\bigg\|^2\bigg) \\
& \bigg\langle \Diamond - \nabla F(x_{k-1}^s), \frac{\sigma_\gamma(\Diamond)}{\Delta}\bigg\rangle\bigg] \\
\leq{} & \mathbb{E}\bigg[F(x_{k-1}^s) - \frac{1}{2L}\bigg(\bigg\langle \Diamond, \frac{\sigma_\gamma(\Diamond)}{\Delta}\bigg\rangle\bigg)^2 \bigg/ L\bigg\|\frac{\sigma_\gamma(\Diamond)}{\Delta}\bigg\|^2 \\
& + \frac{1}{2L}\bigg(\bigg|\bigg\langle \Diamond, \frac{\sigma_\gamma(\Diamond)}{\Delta}\bigg\rangle\bigg| \bigg/ L\bigg\|\frac{\sigma_\gamma(\Diamond)}{\Delta}\bigg\|^2\bigg)\|\Diamond - \nabla F(x_{k-1}^s)\|\bigg\|\frac{\sigma_\gamma(\Diamond)}{\Delta}\bigg\|\bigg] \\
={} & \mathbb{E}\bigg[F(x_{k-1}^s) - \frac{1}{2L}\bigg(\bigg\langle \Diamond, \frac{\sigma_\gamma(\Diamond)}{\Delta}\bigg\rangle\bigg)^2 \bigg/ L\bigg\|\frac{\sigma_\gamma(\Diamond)}{\Delta}\bigg\|^2 \\
& + \frac{1}{2L}\bigg(\bigg|\bigg\langle \Diamond, \frac{\sigma_\gamma(\Diamond)}{\Delta}\bigg\rangle\bigg| \bigg/ L\bigg\|\frac{\sigma_\gamma(\Diamond)}{\Delta}\bigg\|\bigg)\|\Diamond - \nabla F(x_{k-1}^s)\|\bigg] \\
\leq{} & \mathbb{E}\bigg[F(x_{k-1}^s) - \frac{1}{2L}\bigg(\bigg\langle \Diamond, \frac{\sigma_\gamma(\Diamond)}{\Delta}\bigg\rangle\bigg)^2 \bigg/ L\bigg\|\frac{\sigma_\gamma(\Diamond)}{\Delta}\bigg\|^2 \\
& + \frac{\theta}{2L}\bigg(\bigg\langle \Diamond, \frac{\sigma_\gamma(\Diamond)}{\Delta}\bigg\rangle\bigg)^2 \bigg/ \bigg\|\frac{\sigma_\gamma(\Diamond)}{\Delta}\bigg\|^2 + \frac{1}{2L\theta}\cdot\|\Diamond - \nabla F(x_{k-1}^s)\|^2\bigg] \\
={} & \mathbb{E}\bigg[F(x_{k-1}^s) - \frac{1-\theta}{2L}\bigg(\bigg\langle \Diamond, \frac{\sigma_\gamma(\Diamond)}{\Delta}\bigg\rangle\bigg)^2 \bigg/ \bigg\|\frac{\sigma_\gamma(\Diamond)}{\Delta}\bigg\|^2 + \frac{1}{2L\theta}\cdot\|\Diamond - \nabla F(x_{k-1}^s)\|^2\bigg] \\
={} & \mathbb{E}\bigg[F(x_{k-1}^s) - \frac{1-\theta}{2L}\frac{|\langle \Diamond, \sigma_\gamma(\Diamond)\rangle|^2}{\|\sigma_\gamma(\Diamond)\|^2} + \frac{1}{2L\theta}\cdot\|\Diamond - \nabla F(x_{k-1}^s)\|^2\bigg],
\end{aligned}
\tag{17}
$$

where the second inequality uses the Cauchy-Schwartz inequality and the third inequality employs the fact $2ab \leq \theta a^2 + \frac{1}{\theta}b^2$ ($\theta > 0$).

According to the Hölder inequality, we conclude that

$$
\begin{aligned}
\|\sigma_\gamma(\Diamond)\|^2 &= \sum_{i=1}^d |(\Diamond)_i|^{2\gamma} \\
&\leq \bigg(\sum_{i=1}^d \mathbf{1}^p\bigg)^{\frac{1}{p}} \bigg(\sum_{i=1}^d (|(\Diamond)_i|^{2\gamma})^q\bigg)^{\frac{1}{q}} \\
&= \|\mathbf{1}\|_p \bigg(\sum_{i=1}^d |(\Diamond)_i|^{1+\gamma}\bigg)^{\frac{2\gamma}{1+\gamma}},
\end{aligned}
\tag{18}
$$

where $\gamma$ is belonging to the interval $(0, 1)$ with $p = \frac{1+\gamma}{1-\gamma}$ and $q = \frac{1+\gamma}{2\gamma}$.

Further, we have the following quantity

$$
\frac{|\langle \Diamond, \sigma_\gamma(\Diamond)\rangle|^2}{\|\sigma_\gamma(\Diamond)\|^2} \geq \frac{\bigg(\sum_{i=1}^d |(\Diamond)_i|^{1+\gamma}\bigg)^2}{\|\mathbf{1}\|_p \bigg(\sum_{i=1}^d |(\Diamond)_i|^{1+\gamma}\bigg)^{\frac{2\gamma}{1+\gamma}}} = \frac{\|\Diamond\|_{1+\gamma}^2}{\|\mathbf{1}\|_p}.
\tag{19}
$$

The results in (20) and (19) infer the following inequality

$$\mathbb{E}[F(x_k^s)] \leq \mathbb{E}\left[F(x_{k-1}^s) - \frac{1}{2L}\frac{\|\Diamond\|_{1+\gamma}^2}{\|\mathbf{1}\|_p} + \frac{1}{2L\theta}\|\Diamond - \nabla F(x_{k-1}^s)\|^2\right]$$

$$\leq \mathbb{E}\left[F(x_{k-1}^s) - \frac{1-\theta}{2L\|\mathbf{1}\|_p}\left(\frac{1}{2}(1-\alpha_1\beta_1)^2\|\nabla F_{n_k}(x_{k-1}^s)\|_{1+\gamma}^2\right.\right.$$

$$- \|(1-\alpha_1\beta_1)\nabla F_{n_k}(\tilde{x}) - \alpha_1\beta_1 G_{k-1}^s - (1-\alpha_1\beta_1)g^s\|_{1+\gamma}^2\Big)$$

$$+ \frac{1}{2L\theta}\left(4\alpha_1^2\beta_1^2\|G_{k-1}^s\|^2 + \frac{8(1-\alpha_1\beta_1)^2\sigma^2}{B} + 4\alpha_1^2\beta_1^2 \cdot \|\nabla F(x_{k-1}^s)\|^2\right)\bigg]$$

$$= \mathbb{E}\left[F(x_{k-1}^s) - \frac{1-\theta}{4L\|\mathbf{1}\|_p}(1-\alpha_1\beta_1)^2\|\nabla F(x_{k-1}^s)\|_{1+\gamma}^2\right.$$

$$+ \frac{1-\theta}{2L\|\mathbf{1}\|_p}\|(1-\alpha_1\beta_1\nabla F_{n_k}(\tilde{x}) - \alpha_1\beta_1 G_{k-1}^s) - (1-\alpha_1\beta_1)g^s\|_{1+\gamma}^2 + \frac{4\alpha_1^2\beta_1^2}{2L\theta}\|G_{k-1}^s\|^2$$

$$+ \frac{4(1-\alpha_1\beta_1)^2\sigma^2}{LB\theta} + \frac{2\alpha_1^2\beta_1^2\|\nabla F(x_{k-1}^s)\|^2}{L\theta}\bigg]. \tag{20}$$

where the second inequality uses Lemma 2 and Lemma 3.

To satisfy the above inequality, we just need to satisfy the following quantity

$$\mathbb{E}[F(x_k^s)] \leq \mathbb{E}\left[F(x_{k-1}^s) - \frac{(1-\theta)(1-\alpha_1\beta_1)^2}{4L\|\mathbf{1}\|_p}\|\nabla F_{n_k}(x_{k-1}^s)\|_{1+\gamma}^2 + \frac{4(1-\alpha_1\beta_1)^2\sigma^2}{LB\theta}\right]. \tag{21}$$

Telescoping the inequality (21) over $k = 1, \ldots, \mathfrak{K}$, we get

$$\mathbb{E}[F(x_\mathfrak{K}^s)] \leq \mathbb{E}\left[F(x_0^s) - \frac{(1-\theta)(1-\alpha_1\beta_1)^2}{4L\|\mathbf{1}\|_p}\sum_{k=0}^{\mathfrak{K}}\|\nabla F_{n_k}(x_{k-1}^s)\|_{1+\gamma}^2 + \frac{4(1-\alpha_1\beta_1)^2\sigma^2(\mathfrak{K}+1)}{LB\theta}\right]. \tag{22}$$

Since $\tilde{x}^s = x_\mathfrak{K}^s$ and $\tilde{x}^{s-1} = x_0^s$ (shown in Algorithm 3), we further get

$$\mathbb{E}[F(\tilde{x}^s)] \leq \mathbb{E}\left[F(\tilde{x}^{s-1}) - \frac{(1-\theta)(1-\alpha_1\beta_1)^2}{4L\|\mathbf{1}\|_p}\sum_{k=1}^{\mathfrak{K}}\|\nabla F_{n_k}(x_{k-1}^s)\|_{1+\gamma}^2 + \frac{4(1-\alpha_1\beta_1)^2\sigma^2(\mathfrak{K}+1)}{LB\theta}\right]. \tag{23}$$

Through summing (23) over $s = 1, \ldots, \mathfrak{S}$, we obtain

$$\mathbb{E}\left[\sum_{s=1}^{\mathfrak{S}}\sum_{k=1}^{\mathfrak{K}}\|\nabla F_{n_k}(x_{k-1}^s)\|_{1+\gamma}^2\right] \leq \frac{4L\|\mathbf{1}\|_p}{(1-\theta)(1-\alpha_1\beta_1)^2}[F(\tilde{x}_0) - F(\tilde{x}_\mathfrak{K})] + \frac{16\|\mathbf{1}\|_p\sigma^2(\mathfrak{K}+1)(\mathfrak{S}+1)}{(1-\theta)B\theta}$$

$$\leq \frac{4L\|\mathbf{1}\|_p}{(1-\theta)(1-\alpha_1\beta_1)^2}[F(\tilde{x}_0) - F(x_*)] + \frac{16\|\mathbf{1}\|_p\sigma^2(\mathfrak{K}+1)(\mathfrak{S}+1)}{(1-\theta)B\theta}, \tag{24}$$

where the second inequality uses the fact $x_* = \arg\min F(x)$.

Finally, dividing $T$ on both sides of the inequality (24), the desired results is got.

$$\square$$

To acquire an $\varepsilon$-approximate stationary point, i.e., $\mathbb{E}\left[\frac{1}{T}\sum_{s=1}^{\mathfrak{S}}\sum_{k=1}^{\mathfrak{K}}\|\nabla F_{n_k}(x_{k-1}^s)\|_{1+\gamma}^2\right] \leq \varepsilon$, we just satisfy $\frac{4L\|\mathbf{1}\|_p}{(1-\theta)T(1-\alpha_1\beta_1)^2}[F(\tilde{x}_0) - F(x_*)] + \frac{16\|\mathbf{1}\|_p\sigma^2}{(1-\theta)B\theta} \leq \varepsilon$. In particular, considering $B = O(T)$, ADA-PSGD in order

to obtain an $\varepsilon$-approximate stationary point requires $T = O\left(\frac{4\theta\|\mathbf{1}\|_p C + 16\|\mathbf{1}\|_p(1-\alpha_1\beta_1)\sigma^2}{(1-\theta)(1-\alpha_1\beta_1)\theta\varepsilon}\right)$, where we set $C = F(\tilde{x}_0) - F(x_*)$. Further, consider $\mathfrak{K} = o(n)$ and since the total iteration numbers $T$ has been some multiple of $\mathfrak{K}$, therefore, the overall gradient complexity of ADA-PSGD for nonconvex case is $O\left(n + \frac{L^2\|\mathbf{1}\|_p^2}{(1-\alpha_1\beta_1)\varepsilon^2}\right)$.

Note that, for nonconvex optimization problems, earlier studies extended the existing variance-reduced frameworks, attaining the first speeds of order $O\left(n + \frac{n^{2/3}}{\varepsilon^2}\right)$ (Reddi et al., 2016; Zhou et al., 2018). The most recent nonconvex stochastic gradient-based algorithms with variance-reduced techniques approximate this gap and acquire the optimal gradient complexity of $\left(n + \frac{n^{1/2}}{\varepsilon^2}\right)$ (Li et al., 2021b; Pham et al., 2020). With the SVRG-gradient estimator, Yang (2023a) proved the powered stochastic gradient-based method required $O\left(n + \frac{L^2\|\mathbf{1}\|_p^2}{\varepsilon^2}\right)$ gradient computations to acquire an $\varepsilon$-approximate stationary point for nonconvex case. Comparing these results, we conclude that the complexity of our ADA-PSGD method matches that of state-of-the-art stochastic gradient-based algorithms.

## 6  Numerical Evaluation

This section empirically demonstrates the efficacy of ADA-PSGD by comparing it with state-of-the-art stochastic gradient-based algorithms. More specifically, we conduct numerical experiments on two common machine learning tasks, logistic regression (LR) with $\ell_2$-norm regularization and the squared hinge loss support vector machine (SVM) with $\ell_2$-norm regularization:

$$(\textbf{LR}) \quad \min_{x\in\mathbb{R}^d} F(x) = \frac{1}{n}\sum_{i=1}^{n}\log(1 + \exp(-b_i a_i^T x)) + \frac{\lambda}{2}\|x\|^2. \tag{25}$$

$$(\textbf{SVM}) \quad \min_{x\in\mathbb{R}^d} F(x) = \frac{1}{n}\sum_{i=1}^{n}\left(\left[1 - b_i a_i^T x\right]_+\right)^2 + \frac{\lambda}{2}\|x\|^2, \tag{26}$$

where $\{a_i, b_i\}_{i=1}^{n} \in \mathbb{R}^d \times \{+1, -1\}$ is a set of training datasets and $\lambda = 0.01$ denotes a regularization parameter. In addition, to significantly confirm the efficacy of our ADA-PSGD algorithm, we conduct numerical experiments on the nonconvex LR problem:

$$\min_{x\in\mathbb{R}^d} F(x) = \frac{1}{n}\sum_{i=1}^{n}\log(1 + \exp(-b_i a_i^T x)) + \tilde{\lambda} r(x), \tag{27}$$

where $r(x) = \sum_{i=1}^{d} \frac{x_i^2}{1+x_i^2}$ denotes a non-convex regularizer. In the nonconvex LR problem, we adopt $\tilde{\lambda} = 0.1$ for different datasets.

We have performed experiments on four classification datasets, $a8a$, $covtype$, $w8a$, $MNIST$, $CIFAR-10$, and $ijcnn1$, where they are coming from LIBSVM website (Chang & Lin, 2011). Detailed information of these datasets is provided in Table 1.

Table 1: The detailed Information of benchmark datasets

| Data set | Sample size ($n$) | Dimension ($d$) |
|---|---|---|
| $a8a$ | 22,696 | 123 |
| $covtype$ | 581,012 | 54 |
| $ijcnn1$ | 49,990 | 22 |
| $w8a$ | 49,749 | 300 |
| $CIFAR-10$ | 60,000 | 1,024 |
| $MNIST$ | 60,000 | 784 |

## 6.1 Comparison ADA-PSGD with Other Methods

This section compares our ADA-PSGD algorithm with several preferred algorithms: Adam (Kingma, 2014), AdaGrad (Duchi et al., 2011), RMSProp (Tieleman et al., 2012), AMSGRAD (Reddi et al., 2018), SVRG (Johnson & Zhang, 2013), pbVRGD-HD (Yang & Li, 2024), and PB-SVRGE-RSBB (Yang, 2023a). Specifically, Adam, AdaGrad, RMSProp, and AMSGRAD are four classical adaptive gradient methods. pbVRGD-HD and PB-SVRGE-RSBB are two advanced powered stochastic gradient-based methods, where the former used the hypergradient descent technique to compute the learning rate and the latter used the BB-like technique to compute the learning rate.

For Adam and AMSGRAD, we adopt $\beta_1 = 0.9$ and $\beta_2 = 0.999$ for different datasets. For RMSProp, we adopt $\beta = 0.9$ on all datasets. For pbVRGD-HD, we set $\gamma = 0.9$, $\alpha = 0.01$ and $\beta = 0.01$ for different datasets. For PB-SVRGE-RSBB, we also set $\gamma = 0.9$ on all datasets. For our ADA-PSGD algorithm, we set $\gamma = 0.9$, $\alpha_1 = 0.9$, $\alpha_2 = 0.9$, $\beta_1 = 0.1$, $\beta_2 = 0.9$, and $\epsilon = 1$ for different datasets. In addition, on the convex LR problem, we set $\eta = 0.04$ on $a8a$, $ijcnn1$, and $covtype$, and $\eta = 0.06$ on $w8a$ for our ADA-PSGD algorithm. In contrast, on the SVM problem, we also take $\eta = 0.04$ on $a8a$, $ijcnn1$, and $covtype$, but $\eta = 0.1$ on $w8a$ for ADA-PSGD.

Figures 1(a), 1(b), 1(c), and 1(d) show the comparison results among different methods on LR model. In contrast, Figures 1(e), 1(f), 1(g), and 1(h) display the comparison results among different methods on SVM model. All subfigures in Figure 1 clearly demonstrate that our ADA-PSGD method outperforms state-of-the-art optimization algorithms. Particularly, the comparison results among ADA-PSGD, pbVRGD-HD, and PB-SVRGE-RSBB confirm the efficacy of adaptive learning rates in powered stochastic gradient-based algorithms. Also, observed from Figure 1, ADA-PSGD converges faster than Adam, AdaGrad, RMSProp, and AMSGRAD, which confirms the positive impact of the Powerball technique in improving adaptive gradient methods.

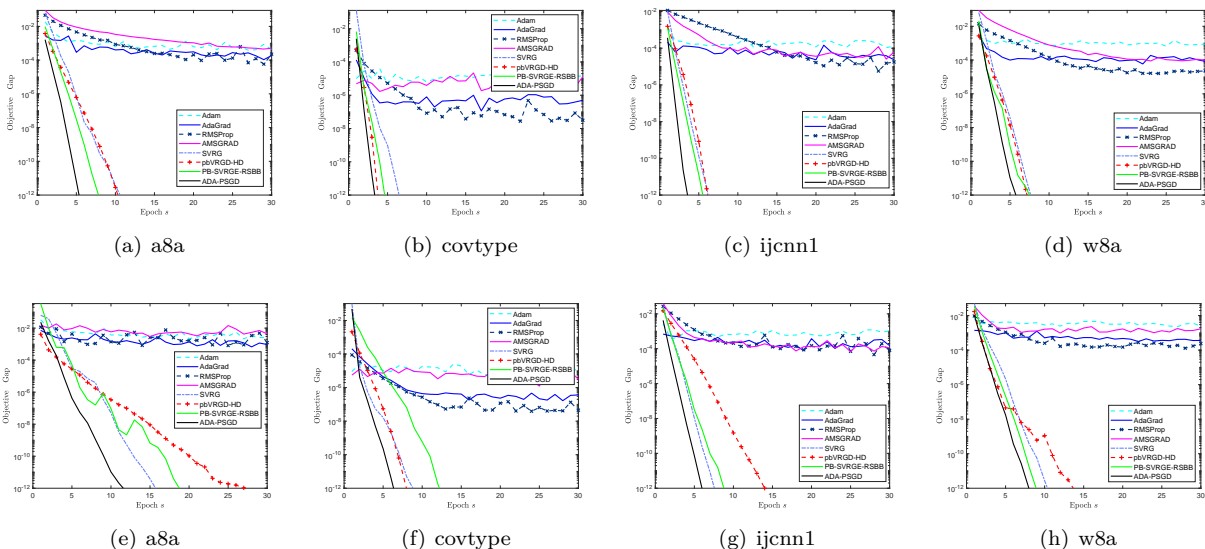

|  |  |  |  |
|---|---|---|---|
| (a) a8a | (b) covtype | (c) ijcnn1 | (d) w8a |
| (e) a8a | (f) covtype | (g) ijcnn1 | (h) w8a |

Figure 1: Comparison of the considered algorithms on the LR model (top line) and the squared hinge loss SVM model (bottom line) with $a8a$ (the first column), $covtype$ (the second column), $ijcnn1$ (the third column), and $w8a$ (the forth column).

Further, we discuss the numerical behavior of ADA-PSGD on the non-convex LR problem in Fig. 2. Note that we also compare our ADA-PSGD algorithm with Adam, AdaGrad, RMSProp, AMSGRAD, SVRG, pbVRGD-HD, and PB-SVRGE-RSBB by performing them on $CIFAR-10$ and $MNIST$. In addition, pbSGD (Yuan et al., 2019) and PB-SGD-RSBB (Yang, 2023a) are also provided as benchmark optimization algorithms. The parameter settings of these algorithms are similar to the experiments on LR and SVM. It

is observed from Fig. 2 that ADA-PSGD still achieves better performance than state-of-the-art stochastic optimization algorithms on the nonconvex LR problem, which further demonstrates the positive role of the Powerball technique in enhancing adaptive gradient methods.

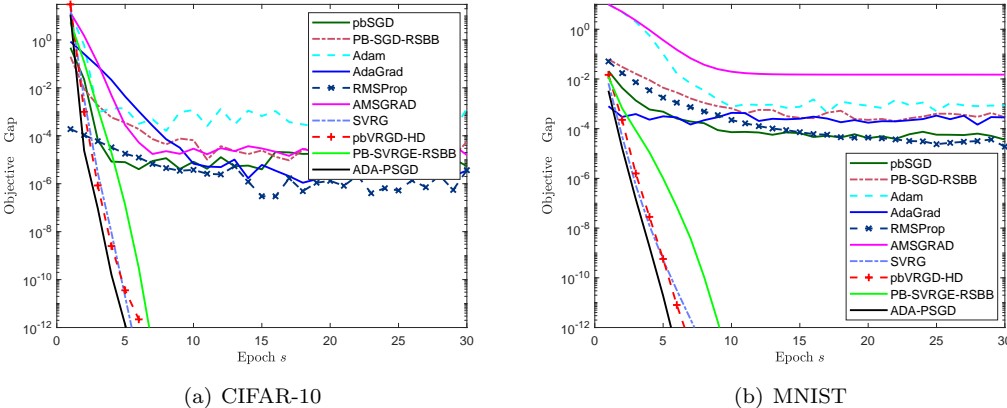

(a) CIFAR-10                                                           (b) MNIST

Figure 2: Comparison of the considered algorithms on the nonconvex LR model with $CIFAR-10$ (left) and $MNIST$ (right).

## 6.2 The Choice of Different Hyper-parameters

Algorithm 3 shows that there are too many hyper-parameters (e.g., $\alpha_1$, $\alpha_1$, $\beta_1$, $\beta_2$, $\gamma$, etc.) that needed to be tuned when performing ADA-PSGD. In order to better comprehend our ADA-OSGD method, this part discusses the numerical behavior of ADA-PSGD with different hyper-parameters.

Figure 3 explores the performance of ADA-PSGD with $\alpha_1$, where we select $\alpha_1$ from {0, 0.3, 0.5, 0.7, 0.9, 1}. Additionally, when executing ADA-PSGD, we uniformly set $\alpha_2 = 0.9$, $\beta_1 = 0.9$, $\beta_2 = 0.9$, and $\epsilon = 1$ on different datasets. More specifically, Figures 3(a), 3(b), 3(c), and 3(d) conduct experiments on LR model, and Figures 3(e), 3(f), 3(g), and 3(h) conduct experiments on SVM.

All subfigures in Figure 3 clearly demonstrate the robustness of ADA-PSGD to the hyper-parameter $\alpha_1$. Actually, when setting $\alpha_1 = 0$, ADA-PSGD turns out to be the powered stochastic gradient-based algorithm with the RMSProp-like method. In contrast, when setting $\alpha_1 = 1$, ADA-PSGD turns out to be the powered stochastic gradient-based algorithm with the Adam-like method.

Figure 4 discusses the numerical behavior of ADA-PSGD with $\beta_1$, where we choose $\beta_1$ from {0, 0.1, 0.3, 0.5, 0.7. 0.9}. In addition, ADA-PSGD works with $\alpha_1 = 0.4$, $\alpha_2 = 0.4$, $\beta_2 = 0.9$, and $\epsilon = 1$ for different datasets. Similarly, Figures 4(a), 4(b), 4(c), and 4(d) show the numerical results of ADA-PSGD on the LR model, while Figures 4(e), 4(f), 4(g), and 4(h) display the numerical results of ADA-PSGD on the SVM model.

Observe from Figure 4, ADA-PSGD is also robust to the hyper-parameter $\beta_1$ on different datasets. It is not difficult to empirically verify the robustness of ADA-PSGD to other crucial hyper-parameters.

In this segment, we further discuss the effect of $\gamma$ in ADA-PSGD by conducting experiments on the LR and SVM models respectively, where $\gamma$ is chosen from {0, 0.3, 0.5, 0.7, 0.9, 1}. More specifically, the numerical results of AD-PSGD with different $\gamma$ are plotted in Figure 5. Note that, when $\gamma = 0$, ADA-PSGD can be regarded as signed powered stochastic gradient-based algorithms with adaptive learning rates. In contrast, when $\gamma = 1$, ADA-PSGD can be viewed as the SVRG-like algorithm with different adaptive learning rates.

Figures 5(a), 5(b), 5(c), and 5(d) perform ADA-PSGD on the LR model, while Figures 5(e), 5(f), 5(g), and 5(h) run ADA-PSGD on the SVM model. Figure 5 shows that a slightly larger $\gamma$ results in a better performance of ADA-PSGD. However, Figure 5 indicates that on most datasets, $\gamma = 0.7$ and $\gamma = 0.9$ make our ADA-PSGD method perform better than the case $\gamma = 1$, which validate the efficacy of the Powerball technique in enhancing stochastic gradient-based algorithms.

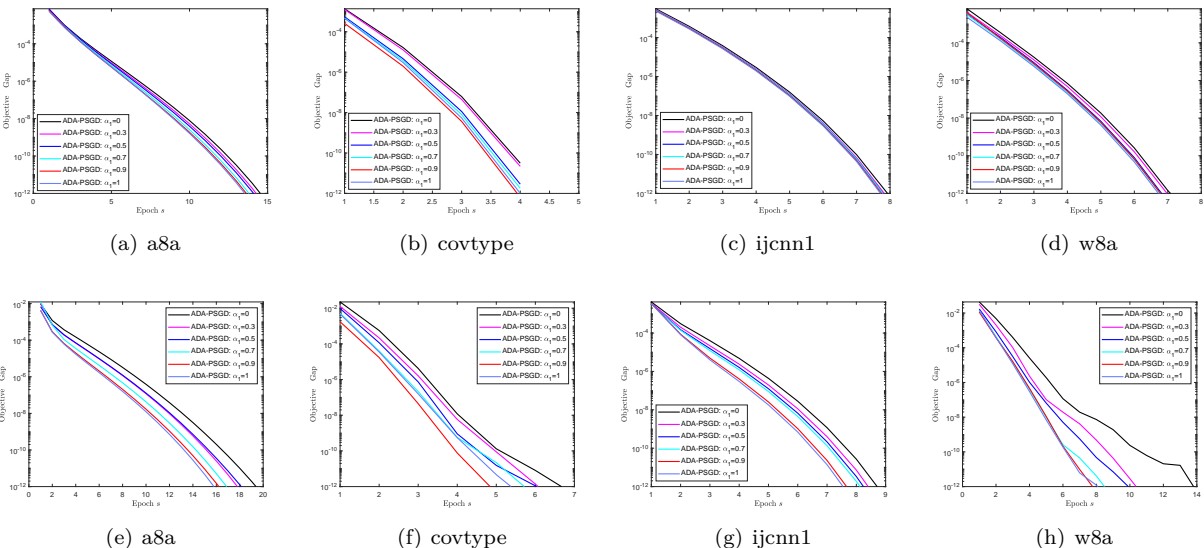

Figure 3: The objective gap, $F(\tilde{x}^s) - F(x_*)$, reported by ADA-PSGD with different $\alpha_1$ on the LR model (top line) and the squared hinge loss SVM model (bottom line) with $a8a$ (the first column), $covtype$ (the second column), $ijcnn1$ (the third column), and $w8a$ (the forth column), where $\alpha_1$ is chosen from $\{0, 0.3, 0.5, 0.7, 0.9, 1\}$.

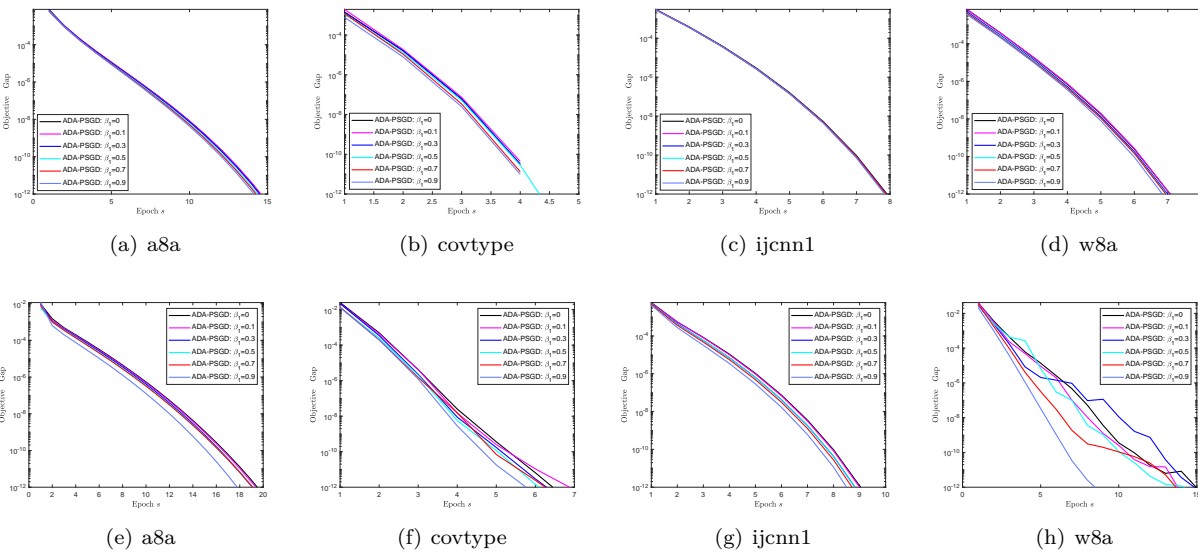

Figure 4: The objective gap, $F(\tilde{x}^s) - F(x_*)$, reported by ADA-PSGD with different $\beta_1$ on the LR model (top line) and the squared hinge loss SVM model (bottom line) with $a8a$ (the first column), $covtype$ (the second column), $ijcnn1$ (the third column), and $w8a$ (the forth column), where $\beta_1$ is chosen from $\{0, 0.1, 0.3, 0.5, 0.7, 0.9\}$.

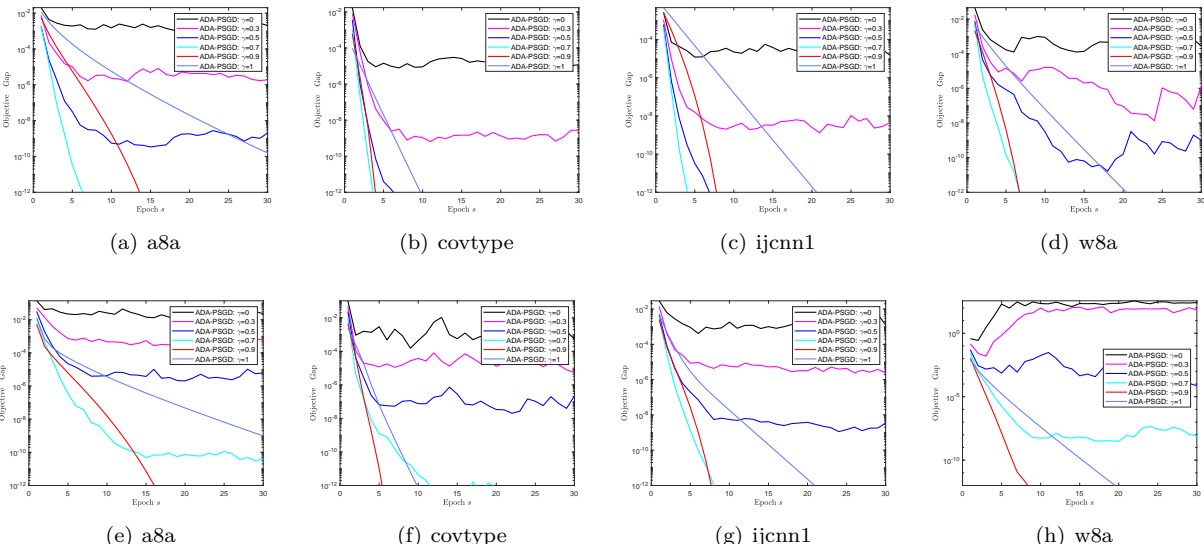

Figure 5: The objective gap, $F(\tilde{x}^s) - F(x_*)$, reported by ADA-PSGD with different $\gamma$ on the LR model (top line) and the squared hinge loss SVM model (bottom line) with $a8a$ (the first column), $covtype$ (the second column), $ijcnn1$ (the third column), and $w8a$ (the forth column), where $\gamma$ is chosen from $\{0, 0.3, 0.5, 0.7, 0.9, 1\}$.

## 7 Conclusion

To comprehend the role of adaptive learning rates in powered stochastic gradient-based algorithms, this work equipped a more generalized adaptive learning rate into powered stochastic gradient-based algorithms, leading to a novel adaptive powered stochastic gradient-based algorithm, named ADA-PSGD. We clearly showed numerous connections of our ADA-PSGD to existing adaptive gradient methods, e.g., Adam, NAdam, RMSProp. A theoretical guarantee of ADA-PSGD for nonconvex optimization problems was established. We proved a faster convergence rate of ADA-PSGD and pointed out that the complexity of our ADA-PSGD method matches that of state-of-the-art stochastic gradient-based algorithms with the magnitude $O\left(\varepsilon^{-2}\right)$. At last, we empirically demonstrated that our ADA-PSGD method leaded to greatly improved training in different machine learning tasks. Moreover, numerical results validated the robustness of ADA-PSGD to crucial hyper-parameters. We hope that all theoretical and empirical advantages of ADA-PSGD will spur interest from both researchers and practitioners.

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
