# OpenReview forum: "Towards Understanding the Role of Adaptive Learning Rates in Powered Stochastic Gradient Descent"
_TMLR — Rejected by TMLR_

### Review · Reviewer_gvmZ · 2024-10-16

**Summary Of Contributions:**

The paper explores how various adaptive learning rate algorithms (ALR) such as Adam, and RMSProp can interact with powered stochastic gradient methods, such as Powerball techniques. The authors introduce ADA-PSGD, a generalized algorithm that incorporates both Powerball methods and ALRs, aiming to combine their strengths. The algorithm is theoretically analyzed, demonstrating faster convergence and improved empirical performance over standard gradient-based algorithms.

**Audience:**

Yes

**Broader Impact Concerns:**

In my opinion, the proposed new algorithm will positively impact society. For example, faster ALR methods with Powerball methods will help reduce the energy consumption from Large Language Model training.

**Claims And Evidence:**

Yes

**Requested Changes:**

1. Enrich the proof scheme for the main Theorem.
2. Conduct a simple deep-learning benchmark for the proposed algorithms.

**Strengths And Weaknesses:**

Strengths:
1. Introduction of ADA-PSGD, a new algorithmic framework that unifies ALR and Powerball methods.
2. A thorough theoretical analysis for ADA-PSGD, showing the computational gradient complexity of $\mathcal{O}(n + \frac{L^2}{\epsilon^2})$, which matches the complexity of state-of-the-art methods.
3. Comprehensive numerical experiments for different algorithms and diverse datasets. This further illustrates the effectivity and robustness of ADA-PSGD compared to classical ALR and Powerball methods.

Weaknesses:
1. Regarding the writing strategy, the proof of the main Theorem is too short (now almost skipped), which might be unfriendly for readers unfamiliar with this area and challenging for reviewers to check the proof's correctness. I cannot directly judge the Theorem's correctness from Lemma 1 to 3.
2. In equations (2) and (3), the updating rule of ADAM and RMSProp is confusing: instead of $\sum_{i=1}^t \sum_{i=1}^t \beta_2^{t-i} g_t^2$, we should have $\sum_{i=1}^t \beta_2^{t-i} g_i^2$?
3. In the numerical experiment part, the authors tested algorithms on logistic regression (LR) and SVM, which are both essentially convex problems. It would be more convincing if the authors could test their algorithms for real-world deep learning problems (no need to be too large-scale, just a demonstration of principle).

---

> ### Author Response · Authors · 2024-10-28
> **Answer to Reviewer gvmZ**
>
> Thank you for reviewing our paper and giving us your advice. Your suggestions are really helpful. Our responses to your concerns are shown below:
>
> Question 1: Enrich the proof scheme for the main Theorem.
>
> Answer: We have shown the full proof of main theorem after Theorem 1. Moreover, the typos had been remedied, such as in the update scheme of Adam and RMSProp, $\psi_t(g_1, \ldots, g_t)$ should be $\psi_t(g_1, \ldots, g_t)=(1-\beta_2) \sum_{i=1}^t \beta_2^{t-i}g_i^2$.
>
> Question 2: Conduct a simple deep-learning benchmark for the proposed algorithms.
>
> Answer: To significantly confirm the effectiveness of the resulting algorithms, we perform our ADA-PSGD algorithm on the nonconvex optimization problem (a.k.a. the nonconvex LR problem). Moreover, we compare ADA-PSGD with state-of-the-art stochastic optimization algorithms on more datasets, such as $CIFAR-10$ and $MNIST$, including  Adam,
> AdaGrad, RMSProp, AMSGRAD, SVRG, pbVRGD-HD, and PB-SVRGE-RSBB. Moreover, pbSGD and PB-SGD-RSBB are also provided as benchmark optimization algorithms. The comparative results on the nonconvex LR problem in Fig. 2 further shows the superiority of ADA-PSGD, which validates the positive effect of the Powerball technique in improving adaptive gradient methods. Future work will report more success of ADA-PSGD on different practical applications.

---

### Review · Reviewer_LnkF · 2024-10-19

**Summary Of Contributions:**

This paper introduces a new algorithm, ADA-PSGD (Algorithm 3 in the paper), that is an adaptive extension of Powered Stochastic Gradient Descent that combines variance-reduction via an SVRG-style estimator, AdaGrad-style adaptive learning rates, and momentum.

**Audience:**

Yes

**Broader Impact Concerns:**

Not applicable.

**Claims And Evidence:**

No

**Requested Changes:**

- (Critical to my recommendation) Please provide the full proof of Theorem 1, make the notation clearer. Provide the choice of stepsizes.
-  (Critical to my recommendation) Why does Theorem 1 not hold for $B=1$ unlike other variance-reduced methods?
-  (Critical to my recommendation) Please make it more clear what exactly we're using adaptive learning rates here for, i.e. what we are adapting to.

**Strengths And Weaknesses:**

There are several issues with this paper:

- (Non-convergence for constant minibatch size) The main weakness of this work is that Theorem 1 does not actually show convergence at all for finite minibatch size that does not vary with \( T \), whereas several other papers studying adaptive variance-reduced methods (e.g. AdaSPIDER) do not. There is no reason why we should not expect an algorithm in this setting to be convergent as well, and the discussion of Theorem 1 compared to prior work ignores.
- (Missing proof and term not clear) First, where is the actual proof of Theorem 1? I don't see how Theorem 1 follows directly from Lemmas 2 and 3. There are also some missing proof details (for example, for the condition ||a||^{2}_{1+\gamma} \geq \|b\|_{1+\gamma}^{2} - \| b-a \|_{1+\gamma}^2). The vector \( 1 \) is used in Theorem 1, but its dimension is never explicitly stated. And in fact, it should be easy to compute the \( p \)-norm of this vector anyway, which would make it easier to parse the theory.
- (Adaptivity unclear in this setting). There are several really unclear points, how is this method adaptive? What is it adapting to exactly? For example, AdaGrad-Norm removes the need for knowing the smoothness constant of the problem, and removes the need to know the variance \( \sigma \) in the non-convex setting. But in the finite-sum setting, with variance reduction, we don't need to know \( \sigma \) anyway. This is made worse because we are not actually given the choices of \( \eta \) and \( \alpha_2 \) that give the rate in Theorem 1.

Based on these issues, I cannot recommend the paper be accepted in its present state.

---

> ### Author Response · Authors · 2024-10-28
> **Answer to Reviewer LnkF**
>
> Thank you for reviewing our paper and giving us your advice. We have revised the draft according to your suggestions and provided explanations for your concerns.
>
> Question 1: Please provide the full proof of Theorem 1, make the notation clearer. Provide the choice of stepsizes.
>
> Answer: The full proof of Theorem 1 has been provided after Theorem 1. Note that in our proof, we consider the base learning rate  (or the base step size pointed out by the reviewer), $\eta=\left\langle\lozenge, \frac{\sigma_\gamma(\lozenge)}{\vartriangle}\right\rangle\bigg/
> L\left\|\frac{\sigma_\gamma(\lozenge)}
> {\vartriangle}\right\|^2>0$, where the token, $\lozenge$, is used to represent $\alpha_1G_k^s+(1-\alpha_1)V_k^s$ and  the token, $\vartriangle$, is used to represent $\sqrt{\alpha_2U_k^s+(1-\alpha_2)(\nabla F_{n_k}(x_{k-1}))^2}+\epsilon$. In addition, the choice of the base learning rate will be discussed in Section 6 (a.k.a. \textbf{Numerical Evaluation}). For instance, on the convex LR problem, we set $\eta=0.04$ on $a8a$, $ijcnn1$, and $covtype$, and $\eta=0.06$ on $w8a$ for our ADA-PSGD algorithm. In contrast, on the SVM problem, we also take $\eta=0.04$ on $a8a$, $ijcnn1$, and $covtype$, but $\eta=0.1$ on $w8a$ for ADA-PSGD.
>
>
> Question 2: Why does Theorem 1 not hold for $B=1$
>  unlike other variance-reduced methods?
>
> Answer: In practice, we can set $B=1$ and make our ADA-PSGD algorithm achieve better performance. Actually, under the case of $B=1$, ADA-PSGD is reduced to the original SVRG method with generalized adaptive gradient techniques and the Powerball function. In other words, Theorem 1 can be held for $B=1$ like other variance-reduced methods.
>
> Question 3: Please make it more clear what exactly we're using adaptive learning rates here for, i.e. what we are adapting to.
>
> Answer: This work is motivated by adaptive gradient methods such as Adam, RMSProp, AdaGrad, etc., where these algorithms are also familiar with adaptive learning rate methods. Specifically, the existing adaptive gradient methods adaptively acquire the learning rate by the form of exponentially decaying average of squared historical gradient values. In this work, we empowered the powered stochastic gradient-based algorithms with the ability to automatically determine the learning rate by employing generalized adaptive gradient methods.

---

### Review · Reviewer_Xn3j · 2024-10-24

**Summary Of Contributions:**

The authors propose an algorithm which incorporates a large family of adaptive learning rate methods containing Adam, RMSProp and NAdam, into the powerball framework.
Note that they minimize $L$-smooth non-convex function $F$ which is a finite sum of $n$ individual losses.
Vanilla powerball methods (Yang 2023) apply the function $\sigma_{\gamma}(\cdot)$ to the gradient before updating iterates, where for each coordinate $\sigma_{\gamma}(a) = sign(a)|a|^\gamma$ for $a\in \mathbb{R}$.

The main contributions of this paper are --
1. **Algorithm** : The proposed algorithm, ADA-PSGD is a direct extension of (Yang 2023) which utilize powerball method with variance reduction and a variable learning rate $\eta$. Instead, the authors combine variance reduction with Adam-style adaptive learning rates.
2. **Theory**: Note that the authors provide an incomplete proof of their theoretical results with several mistakes. If they could provide the complete proofs, then the convergence analysis for this family of adaptive learning rate variance reduced powerball methods would have been a significant result as it would have matched rates of (Yang 2023) who only use a variable learning rate.
3. **Experiments**: On convex functions (regularized logistic regression and SVMs) and simple datasets(a8a, covtype, ijcnn1, w8a), their algorithm outperforms existing baselines and is robust to change in it's hyperparameters.

**References**
- (Yang 2023)  Improved Powered Stochastic Optimization Algorithms for Large-Scale Machine Learning. JMLR.

**Audience:**

Yes

**Claims And Evidence:**

No

**Requested Changes:**

## Requested Changes --
1. **Algorithm**: Provide examples of existing adaptive learning rate algorithms which are obtained for $\alpha_2\neq 1$.
2. **Theory**: Please provide a complete proof of Theorem 1 and answer the additional questions raised in the weaknesses assuming Theorem 1 is true.
3. **Experiments**: Provide analysis of ADA-PSGD and baselines on non-convex problems, simple ones like PL functions, those used in (Yang 2023) or on small neural networks. Include more complicated datasets, for instance, MNIST or CIFAR10. Provide tuning details for baselines and standard deviations in Figure 1.
4. Fix the typos.

**References**
- (Yang 2023)  Improved Powered Stochastic Optimization Algorithms for Large-Scale Machine Learning. JMLR.


Overall, the theoretical contributions are invalidated due to lack of proof and other concerns raised in the weaknesses section. Without the theoretical contributions, the only contributions of the paper are the algorithm and experiments for convex losses on simple datasets, which are not sufficient for this paper to be accepted.
I'm willing to change my assessment only if the authors are able to prove Theorem 1, address all theoretical concerns and provide experiments on non-convex losses.

**Strengths And Weaknesses:**

## Strengths --
1.  **Algorithm**: Combining these two types of algorithms (Variance reduced powerball and adaptive learning rates) is a non-trivial and novel task, and the authors are able to recover updates resembling Adam, RMSProp and NAdam for different choices of parameters.
2. **Experiments** : Their algorithm outperforms baselines, even (Yang 2023) and is robust to change in hyperparameters.

## Weaknesses --

1. **Algorithm**: The parameter $\alpha_2$ in the Algorithm seems to be redundant. In the update step for $x_{k}^s$, the coefficient $\alpha_2$ is used to balance the contribution of $U_k^s$, the running estimate of gradient norm, and the current gradient norm $\|\|\nabla F_{n_k}(x_{k-1}^s)\|\|^2$. However, the update step of $U_{k-1}^s$ has already incorporated the running estimate of gradient norm and the current gradient norm with the coefficient $\beta_2$. Further, in Remark, point (1) on page 5, all hyperparameter values to recover known algorithms set $\alpha_2=1$, which basically removes the contribution of $\alpha_2$.
2. **Theory**: The authors only provide assumptions, key lemmas and the final theorem in Section 5 and do not provide a full proof. While the individual Lemmas are true under the assumptions, it is highly non-trivial to go from the Lemmas to the final theorem. It seems that the authors are following (Yang 2023), as their final theorem statements are similar, however, we cannot directly apply the proof technique of (Yang 2023) for this problem, as the algorithms differ significantly. I request the authors to provide a proof, if they have it ready but missed putting it in the submission. Without the proof, the theoretical results cannot be assumed to be true. Even assuming the theoretical results are true, there are several issues. These are described below.
    - What is $\theta$ in Theorem 1?
    - Why does $\alpha_2$ and $\beta_2$ not appear in the final convergence rate in Theorem 1? Can I set $\alpha_2$ and $\beta_2$ to any arbitrary value?
    - How is $O(n)$ term for number of gradient evaluations obtained in the last paragraph of Section 5? The total number of iterations is $O(\mathfrak{S}(n + \mathfrak{K}B)) = O(\mathfrak{S}n + T^2)$. Even if $\mathfrak{K} = o(n)$ (let $\mathfrak{K} = c n^{1-a}$ for some $a\in (0,1]$ and constant $c>0$) we can have $\mathfrak{S} = \frac{Tn^{a-1}}{c}$ and thus the first term $O(\mathfrak{S}n) = O(Tn^a) = O(\frac{L\|1\|_p n^a}{(1 - \alpha_1\beta_1)\epsilon})$ which is not equal to and in fact can be much larger than $O(n)$. I feel there is a similar problem in the last paragraph of Section 3 page 9 in (Yang 2023a) on which this analysis is based.
    - How does this method compare to the PB-SGD-RSBB algorithm (Yang 2023), considering both ADA-PSGD and PB-SVRGE-RSBB use Adaptive learning rates? Is ADA-PSGD better under certain conditions? The theoretical convergence for PB-SVRGE-RSBB is also faster than PB-SGD (by appropriately setting $b_H$ in (Theorem 3, Yang 2023)) and should therefore be the state-of-the-art powerball method which the authors should compare to.
3. **Experiments**:
    - Although the authors show convergence for non-convex losses, they consider only convex losses in their experiments. In contrast, (Yang 2023) consider a non-convex logistic loss. Therefore, we do not know if the theoretical convergence of ADA-PSGD on non-convex losses actually translates to practice.
    - It is unclear if the hyperparameters for all baseline algorithms have been tuned before experiments, which might make the comparison unfair for the baselines.
    - No standard deviations are plotted.
4. **Typos**: The paper contains several typos, even in the algorithm and the proofs.
    - The algorithm in the update step for $U_k^s$ and $x_k^s$ adds $(\nabla F_{n_k}(x_{k-1}^s))^2$ which does not make sense as this a vector. Ideally, its $\ell_2$ norm should be added.
    - Lemma 1 : LHS should have $\ell_{1+  \gamma}$ norm instead of $\ell_2$ norm.
    - Proof of Lemma 1 :  First and third equations should have $\alpha_1 \beta_1 G_{k-1}^s$ instead of $\alpha_1\beta_2 G_{k-1}^s$.
    - Proof of Lemma 2 : In the second equation, the third term should be $-(1 -\alpha_1\beta_1)\nabla F(x_{k-1}^s)$ instead of $-(1 -\alpha_1\beta_1)\nabla F(\tilde{x})$ and the last inequality should have $\alpha_1^2\beta_1^2\|\|G_{k-1}^s\|\|^2$ instead of $\alpha_1\beta_1\|\|G_{k-1}^s\|\|^2$.

**References**
- (Yang 2023)  Improved Powered Stochastic Optimization Algorithms for Large-Scale Machine Learning. JMLR.

---

> ### Author Response · Authors · 2024-10-28
> **Answer to Reviewer Xn3j**
>
> Question 1: Provide examples of existing adaptive learning rate algorithms...
>
> Answer: On the one hand, the use of $\alpha_2$ can balance the contribution of second moment estimate, $U_k$. On the other hand, when adaptive gradient methods, such as ADAM, NAdam, etc., are instable, it is possible that selecting appropriate $\alpha_1$ and $\alpha_2$ can improve stability by forcing a tighter learning rate bound. For instance, the pre-coordinate learning rate of Adam is bounded by $\eta\cdot\frac{1-\beta_1}{\sqrt{1-\beta_2}}$. In contrast, for convenience, considering $\gamma=1$, the learning rate of our ADA-PSGD algorithm is bounded by $\eta\cdot\frac{\alpha_1(1-\beta_1)}
> {\sqrt{\alpha_2(1-\beta_2)}}$. Actually, most of the existing adaptive gradient methods only consider the second moment estimate in Adam-like algorithms that are $\alpha_2=1$ (a.k.a. $x_k=x_{k-1}-\eta\frac{m_t}{\sqrt{v_t}}$), appearing in our ADA-PSGD algorithm.
>
>
>
> Question 2: Please provide a complete proof of Theorem 1 and answer the additional questions raised in....
>
> Answer: We have shown the full proof of Theorem 1 in the revised version of the PDF manuscript.
>
> Question 3: Provide analysis of ADA-PSGD and baselines on non-convex problems...
>
> Answer: We will show the performance of our ADA-PSGD algorithm on the non-convex logistic regression problem, as shown in (Yang 2023). More specifically, we conduct experiments on $MNIST$ and $CIFAR-10$. Note that we also compare our ADA-PSGD algorithm with  Adam, AdaGrad, etc. Additionally, pbSGD and PB-SGD-RSBB are also provided as benchmark optimization algorithms. It is observed from Fig. 2 that ADA-PSGD still achieves better performance than state-of-the-art stochastic optimization algorithms on the nonconvex LR problem, which further demonstrates the positive role of the Powerball technique in enhancing adaptive gradient methods.
>
> Question 4: Fix the typos.
>
> Answer: We have fixed the typos. For instance, in Lemma 2, we replace $\ell_2$ norm by $\ell_{1+\gamma}$ in the left hand side of the inequality (10). For Lemma 3, in the second equation,  the third term is replaced by $-(1-\alpha_1\beta_1)\nabla F(x_{k-1}^s)$ and in the last inequality, we replace $\alpha_1\beta_1\|G_{k-1}^s\|^2$ by $\alpha_1^2\beta_1^2\|G_{k-1}^s\|^2$. In addition, $ \psi_t(g_1, \ldots, g_t)=(1-\beta_2)\sum_{i=1}^t \sum_{i=1}^t \beta_2^{t-i}g_i^2$ should be  $\psi_t(g_1, \ldots, g_t)=(1-\beta_2)\sum_{i=1}^t \beta_2^{t-i}g_i^2$. In Assumption 1, "Problem equation 1" should be "Problem (1)", to name a few. Note that in Algorithm 3, it is fine to use $(\nabla F_{n_k}(x_{k-1}^s))^2$ when updating $U_k$, since $(\nabla F_{n_k}(x_{k-1}^s))^2$ means that each element in vector $\nabla F_{n_k}(x_{k-1}^s)$ needs to be squared. This is indeed the case, because $U_k$ itself is a vector.
>
>
> Question 2.1: What is $\theta$ in Theorem 1?
>
> Answer: In Theorem 1, $\theta$ is a positive constant, appearing the proof of Theorem 1. We will emphasize this in Theorem 1.
>
> Question 2.2: Why does $\alpha_2$ and $\beta_2$ not appear...
>
> Answer: To finish the proof of Theorem 1, by setting $\eta=\left\langle\lozenge, \frac{\sigma_\gamma(\lozenge)}{\vartriangle}\right\rangle\bigg/
> L\left\|\frac{\sigma_\gamma(\lozenge)}
> {\vartriangle}\right\|^2>0$, where the token, $\lozenge$, is used to represent $\alpha_1G_k^s+(1-\alpha_1)V_k^s$ and  the token, $\vartriangle$, is used to represent $\sqrt{\alpha_2U_k^s+(1-\alpha_2)(\nabla F_{n_k}(x_{k-1}))^2}+\epsilon$, we can reasonably eliminate the term that contains $\alpha_2$ and $\beta_2$. However, it doesn't mean that we can set $\alpha_2$ and $\beta_2$ to be any arbitrary value, where we have emphasized that $\beta_2 \in (0, 1)$ and $\alpha_2 \in [0, 1]$ in the paper.
>
> Question 2.3: How is $O(n)$ term for number of gradient evaluations....
>
> Answer: Considering $\mathfrak{S}=T/\mathfrak{K}$, the total number of iterations is $O(\mathfrak{S}(n+\mathfrak{K}B))
> =O\left(\frac{T}{\mathfrak{K}}(n+\mathfrak{K}B)\right)
> =O\left(\frac{nT}{\mathfrak{K}}+BT\right)$. Note that since $T$ is some multiple of $\mathfrak{K}$ and consider $\mathfrak{K}=o(n)$, we have that the complexity of ADA-PSGD is $O(n+T^2)$, where we adopt $b=O(T)$. This is how we obtain the $O(n)$ term for the number of gradient evaluations obtained in the last paraph of Section 5. Further, under the case $T=O\left(\frac{4\theta\|\textbf{1}\|_pC+16\|\textbf{1}\|_p(1-\alpha_1\beta_1)\sigma^2}
> {(1-\theta)(1-\alpha_1\beta_1)\theta\varepsilon}\right)$, the overall complexity of ADA-PSGD for nonconvex optimization problem is $O\left(n+\frac{L^2\|\textbf{1}\|_p^2}
> {(1-\alpha_1\beta_1)\varepsilon^2}\right)$.
>
>
>
> Question 2.4: How does this method compare to the PB-SGD-RSBB algorithm (Yang 2023)...
>
> Answer: To significantly show the efficacy of ADA-PSGD, we will compare ADA-PSGD with PB-SGD and PB-SGD-RSBB by performing numerical experiments on the nonconvex LR problem. Concretely, the comparison results among different algorithms are presented in Fig. 2.

---

### Decision · Action_Editor_FAeK · 2024-12-10

**Recommendation:** Reject

**Comment:**

This paper presents an extension of the powered stochastic optimization framework of Yang (2023), which aims to recover several existing adaptive optimization algorithms as special cases. The proposed algorithm combines variance reduction with the Adam-style momentum update. Theoretical analysis is provided, and the practical advantage compared to certain baselines is demonstrated through experiments.

The critical weakness of this paper is that the proof of the main theorem is incomplete, and the issue remains after the authors updated the manuscript during the rebuttal. Specifically, there is a significant gap going from Eq (20) to Eq (21), where certain terms in the proof are dropped without justification. This is very well summarized by one of the reviewers' official recommendation:

> Incomplete Proof (Equations (20) and (21)): From Equation 20 to equation 21 in their proof, the authors remove positive terms from the upper bound, the third, fourth and sixth terms in the RHS of Eq 20. Further, second term of Eq 20, which contains
-\mathbb{E}[||\nabla F_{n_k}(x_{k-1})||^2] as the second term of Eq 21. Equation 20 is obtained from Lemmas 2 and 3. Handling the
 positive terms in addition to the difference in the second terms of Eq 21 and Eq 20 should be fairly complicated. The authors, however, ignore them to obtain equation 21, which can be simplified to their final result, Theorem 1. Their proof is incomplete.

The reviewers also raised a number of additional concerns on the technical aspects of the theoretical result, including the use of time-varying $\eta$ in the proof, the requirement of unreasonably large batch size, and the absence of hyperparameters $\alpha_2$ and $\beta_2$ (explicitly) in the bound. The exposition of the paper also needs major improvements, particularly regarding various unclear notations in the theoretical analysis.

Overall, the paper in its current form does not meet the criteria of acceptance.

**Audience:**

Yes.

**Claims And Evidence:**

No. The proof of the main theorem is incomplete. In addition, the result lacks clarity, and suffers from a number of noticeable technical weaknesses.